# The Lipid Receptor G2A (GPR132) Mediates Macrophage Migration in Nerve Injury-Induced Neuropathic Pain

**DOI:** 10.3390/cells9071740

**Published:** 2020-07-21

**Authors:** Tabea Osthues, Béla Zimmer, Vittoria Rimola, Kevin Klann, Karin Schilling, Praveen Mathoor, Carlo Angioni, Andreas Weigert, Gerd Geisslinger, Christian Münch, Klaus Scholich, Marco Sisignano

**Affiliations:** 1Fraunhofer Institute for Molecular Biology and Applied Ecology IME, Branch for Translational Medicine and Pharmacology TMP, Theodor-Stern-Kai 7, 60596 Frankfurt am Main, Germany; Tabea.Osthues@ime.fraunhofer.de (T.O.); geisslinger@em.uni-frankfurt.de (G.G.); scholich@em.uni-frankfurt.de (K.S.); 2Institute of Clinical Pharmacology, *pharmazentrum frankfurt/ZAFES*, University Hospital, Goethe-University, D-60590 Frankfurt am Main, Germany; Zimmer@med.uni-frankfurt.de (B.Z.); rimola@med.uni-frankfurt.de (V.R.); Schilling@med.uni-frankfurt.de (K.S.); angioni@em.uni-frankfurt.de (C.A.); 3Institute of Biochemistry II, Faculty of Medicine, University Hospital, Goethe-University, D-60590 Frankfurt am Main, Germany; klann@em.uni-frankfurt.de (K.M.); ch.muench@em.uni-frankfurt.de (C.M.); 4Institute of Biochemistry I, Faculty of Medicine, Goethe-University, D-60590 Frankfurt am Main, Germany; mathoor@biochem.uni-frankfurt.de (P.M.); weigert@biochem.uni-frankfurt.de (A.W.); 5Cardio-Pulmonary Institute, D-60590 Frankfurt am Main, Germany

**Keywords:** neuropathic pain, 9-HODE, oxidized linoleic acid metabolites, macrophage migration, G2A, GPR132

## Abstract

Nerve injury-induced neuropathic pain is difficult to treat and mechanistically characterized by strong neuroimmune interactions, involving signaling lipids that act via specific G-protein coupled receptors. Here, we investigated the role of the signaling lipid receptor G2A (GPR132) in nerve injury-induced neuropathic pain using the robust spared nerve injury (SNI) mouse model. We found that the concentrations of the G2A agonist 9-HODE (9-Hydroxyoctadecadienoic acid) are strongly increased at the site of nerve injury during neuropathic pain. Moreover, G2A-deficient mice show a strong reduction of mechanical hypersensitivity after nerve injury. This phenotype is accompanied by a massive reduction of invading macrophages and neutrophils in G2A-deficient mice and a strongly reduced release of the proalgesic mediators TNFα, IL-6 and VEGF at the site of injury. Using a global proteome analysis to identify the underlying signaling pathways, we found that G2A activation in macrophages initiates MyD88-PI3K-AKT signaling and transient MMP9 release to trigger cytoskeleton remodeling and migration. We conclude that G2A-deficiency reduces inflammatory responses by decreasing the number of immune cells and the release of proinflammatory cytokines and growth factors at the site of nerve injury. Inhibiting the G2A receptor after nerve injury may reduce immune cell-mediated peripheral sensitization and may thus ameliorate neuropathic pain.

## 1. Introduction

Neuropathic pain is a form of chronic pain induced by lesions, diseases, chemicals or tumor invasion of the somatosensory nervous system [1,2]. Neuropathic pain affects millions of people worldwide and is difficult to treat due to lack of satisfactory medication and undesirable side effects [3,4,5].

Typical symptoms of neuropathic pain are allodynia—pain perception by harmless stimuli, hyperalgesia—increased pain perception by painful stimuli, or spontaneous pain [6,7,8,9,10]. Thereby, the normally high thresholds of peripheral nociceptors to mechanical and thermal stimuli are drastically reduced [9,11,12,13]. This peripheral sensitization is caused by neuroimmune interactions involving migration and infiltration of immune cells to the site of injury and the release of proinflammatory factors in nerve injury-induced neuropathic pain [6,11,14]. This interaction is known to modulate neuronal ion channels, such as the transient receptor potential vanilloid 1 (TRPV1) channel, thereby altering pain perception during neuropathic pain [6,12,13]. In nerve injury-induced neuropathic pain during the process of neuroimmune communication, signaling lipids play an important role and several lipid receptors have been identified as crucial mediators of onset, maintenance and resolution of different pathological pain states [15]. The linoleic acid metabolite 9-hydroxyoctadecadienoic acid (9-HODE) is the ligand of the G-protein coupled receptor G2A (GPR132), which is known to sensitize TRPV1 via G_q_ and activation of protein kinase C (PKC) [15,16,17]. G2A belongs to the group of proton-sensing GPCRs and is expressed in TRPV1-positive primary sensory neurons, but mainly in immune cells [18,19]. In the group of proton-sensing GPCRs, G2A shows the weakest response to acidic pH and seems to be a receptor for signaling lipids rather than for acidic conditions [20,21]. It has also been suggested before that G2A is responsible for the migration of leukocytes, but the underlying signaling pathways are still unclear [22,23,24].

The strong expression of this crucial lipid receptor in both sensory neurons and immune cells led us to the hypothesis that G2A is involved in immune cell migration and peripheral sensitization after nerve injury.

Here, we used the spared nerve injury, an acute neuropathic pain model with a strong inflammatory component, to investigate the role of G2A in neuropathic pain. We show that G2A-deficiency results in reduced mechanical hypersensitivity in vivo and leads to a markedly reduced number of immune cells and inflammatory mediators at the site of nerve injury, indicating a pivotal role of the G2A receptor in the initiation and progression of nerve injury-induced neuropathic pain.

## 2. Materials and Methods

### 2.1. Ethics Statement

All animals involved in the presented experiments were approved by the local Ethics Committees for Animal Research (Darmstadt, Germany) under the permit numbers FK/1046 and FK/1113. The animal experiments were performed according to the recommendations of the Guide of the Care and Use of Laboratory Animals of the National Institutes of Health (Guide for the Care and Use of Laboratory Animals, Washington, DC, USA, 2011). All efforts were made to minimize suffering.

### 2.2. Animals

In all behavioral experiments, wild-type and G2A-deficient mice with C57BL/6NRj background were matched in sex and age (9–16 weeks). Wild-type mice were purchased from commercial breeding companies (Janvier, Le Genest-Saint-Isle, France). The G2A-deficient mice were generated previously in the lab of Owen Witte, University of California (San Francisco) [25] and were bred at MFD diagnostics (Wendelsheim, Germany). During all behavioral experiments, the experimenter was blinded. Preclinical pain experiments were in accordance with the suggestions from the Preclinical Pain Research Consortium for Investigating Safety and Efficacy (PPRECISE) Working Group [26].

### 2.3. Spared Nerve Injury (SNI)

For behavioral experiments, spared nerve injury was performed. After anesthesia of the mice, the sciatic nerve was exposed by blunt dissection on the level of the knee joint. Then, the peroneal and the tibial branches of the sciatic nerve were ligated with 6/0 non-sterile silk thread and cut distally from the ligature [27].

### 2.4. Behavioral Experiments

Determination of mechanical pain thresholds were performed by using a Dynamic Plantar aesthesiometer (Ugo Basile, Comerio, Italy). For this purpose, mice were sitting on an elevated grid in test cages for at least 1 h for accommodation before measurement. For baseline measurements, a steel rod was applied to the plantar side of the hind paw with linear ascending force (0–5 g over 10 s, in 0.5 g/s intervals). The time of the first contact of the steel rod until paw withdraw was measured in seconds (paw withdraw latency). Here, a cut-off time of 20 s was set [28]. After baseline measurements, the SNI surgery was performed followed by further dynamic plantar measurements.

### 2.5. Tissue Isolation

Mice were sacrificed and the sciatic nerve was dissected from injured (ipsilateral) and uninjured sites (contralateral) followed by freezing either in liquid nitrogen for RNA isolation or embedding in tissue tek (Sakura Finetek Europe, Alphen aan den Rijn, Netherlands) for slice preparation. Afterwards, the spinal cord of L1–L4 was dissected and incubated overnight in 20% sucrose solution at 4 °C. Then, the spinal cord was embedded vertically in tissue tek for following serial sections of 12 µm thickness by use of a cryostat (Leica, Frankfurt, Germany).

After dissecting the spinal cord, L4 to L6 DRGs were isolated and shock-frozen in fluid nitrogen for RNA isolation.

### 2.6. qRT-PCR

RNA was isolated from DRGs with a mirVana miRNA Isolation Kit (Applied Biosystems). Afterwards, RNA was quantified with a NanoDrop ND-1000 spectrophotometer (NanoDrop Technologies, Waltham, MA, USA). 400 ng RNA was used for cDNA synthesis. For reverse transcription, the First Strand cDNA Synthesis Kit was used according to the manufacturer’s description). For the following qPCR, a TaqMan^®^ Gene Expression Assay System was used according to the manufacturers description (Thermo Scientific, Waltham, MA, USA) and as previously reported [29]. Afterwards, the qPCR program was conducted with the QuantStudio™ Design and Analysis Software v1.4.3 (Thermo Fisher,) and evaluated using the ΔΔC(T) method, as described previously [30].

### 2.7. Immunohistochemistry

For immunohistochemical staining, spinal cord or nerve tissue was prepared into 12 µm slices. The slices were fixed with 2% PFA in PBS (pH 7.4) for 20 min. After washing the slices for 5 min with PBS, they were permeabilized in PBST (PBS + 0.1% triton X) for 10 min. Then, samples were blocked for 45 min in 3% BSA in PBST. As first antibodies, GFAP (NB300-141, novusbio, Centennial, CO, USA) in 1:1000 dilution for spinal cord or F4/80 (ab6640, abcam, Cambridge, UK) and CD11b (ab133357, abcam) in 1:100 dilution for nerve were used. After incubation over night at 4 °C, the second antibodies, anti-rabbit Cy 3 (C2306-1ML, Sigma, Deisenhofen, Germany) and anti-rat AF488 (ab150157, abcam), were incubated in a dilution of 1:1000 at room temperature for 1 h. Afterwards, the slices were stained with DAPI 1:1000 (6335.1, Carl Roth, Karlsruhe, Germany). Pictures of the stained slices were taken with the fluorescence microscope Observer.Z1 (Carl Zeiss, Oberkochen, Germany) [31].

### 2.8. ELISA and Multiplex Assays

For protein detection, dissected nerve tissues were chopped, spinal cord samples were pottered, and afterwards, DRG, spinal cord and nerve samples were sonicated (2 × 60%, 10 s) in 100–500 µL of, respectively, cell lysis buffer, a mixture of phosphosafe (Merck, Darmstadt, Germany) and protease-inhibitor (Roche Holding AG, Basel, Switzerland) or ice-cold PBS (R&D systems, Minneapolis, MN, USA/Ray Biotech, Peachtree Corners, GA, USA). Afterwards, the samples were centrifuged for 10 min at 10,000 rpm. The supernatant was then used for Bradford (Sigma-Aldrich) measurements, as described previously (Bradford 1976), followed by IL-1β (R&D systems, Minneapolis, Minnesota, USA), IL-6, TNFα (Ray Biotech), NGF (DLDevelop, Wuxi, Jiangsu, China), ELISA or Luminex Multiplex measurements (Invitrogen, Carlsbad, CA, USA). In the Luminex Multiplex measurement, the following cyto- and chemo-kines were measured: FGFβ, GM-CSF, IFN-γ, IL-1α, IL-1β, IL-2, IL-4, IL-5, IL-6, IL-10, IL-12 (p40/p70), IL-13, Il-17, IP-10, KC, MCP-1, MIG, MIP-1α, TNF-α and VEGF. The different ELISAs and the Luminex Multiplex were performed according to the manufacturer’s description. All samples were measured in duplicates. For calculation of the protein concentrations, Graph Pad Prism 7 was used.

### 2.9. Differentiation and Stimulation of Bone Marrow-Derived Macrophages

Mice were sacrificed, and the hind legs were dissected from the body and transferred into PBS. The bones were cleaned from muscle tissue, cut open and then centrifugated (13,000 rpm, 1 min, RT) to isolate the bone marrow. Afterwards, 12.5 mL medium (RPMI 1640—GlutaMAX™-I, Invitrogen, Carlsbad, USA, with 10% FCS, 1% penicillin/streptomycin) was mixed with 0.2 µL/mL M-CSF (100 µg/mL, Preprotech, Hamburg, Germany) and used for resuspension of the bone marrow. Afterwards, the suspension was transferred to a 6-well plate. Cells were incubated over night at 37 °C and cultivated for seven days, as previously described [22].

Then cells were stimulated with 1 µM 9 HODE for 0, 5, 10 and 15 min, or 24 h. Afterwards, cells were harvested with a cell scraper in 50:50 Phosphosafe:cOmplete protease-inhibitor cocktail (100 µL/well). Then, the cells were sonicated twice (60%, 10 s) and total protein amount was determined by Bradford Protein Assay [16,22,32].

### 2.10. Proteome Anaylsis

#### 2.10.1. Sample Preparation for LC-MS^2^

Proteomics were performed as previously described with some adjustments [33]. Lysates were precipitated using 3 volumes of ice-cold methanol, 1 volume Chloroform and 2.5 volumes ddH2O. After centrifugation (14,000× *g*, 45 min, 4 °C), the upper aqueous phase was aspirated, and 3 volumes of ice-cold methanol were added. Samples were mixed and proteins pelleted by centrifugation (14,000× *g*, 5 min, 4 °C). The pellets were washed with ice-cold methanol. Protein pellets were dried at RT and then resuspended in 8 M Urea, 10 mM EPPS pH 8.2 and 1 mM CaCl_2_, followed by protein concentration determination using a µBCA assay (ThermoFisher Scientific, 23235). Samples were then diluted to 2 M urea using digestion buffer (10 mM EPPS pH 8.2, 1 mM CaCl_2_) and incubated with the endoproteinase LysC (Wako Chemicals, Neuss, Germany) at a 1:50 (*w*/*w*) ratio overnight at 37 °C. Then, digestion reactions were diluted to 1 M Urea using digestion buffer and incubated at a 1:100 (*w*/*w*) ratio of trypsin (V5113, Promega, Madison, WI, USA) for 6 h at 37 °C. Digests were acidified to a pH of 2–3 using trifluoroaceticacid (TFA). Peptides were purified with SepPak tC18 columns (WAT054955, Waters Milford, MA, USA) according to the manufacturer’s instructions. Eluates were dried and peptides were resuspended in TMT labeling buffer (0.2 M EPPS pH 8.2, 10% Acetonitrile). Peptide concentration was determined by µBCA. Peptides were mixed with TMT reagents (ThermoFisher Scientific, 90111, A37724, 90061) at 1:2 (*w*/*w*) (2 µg TMT reagent per 1µg peptide). Reactions were incubated (1 h, RT) and subsequently quenched by addition of hydroxylamine to a final concentration of 0.5% (15 min, RT). Samples were pooled in equimolar ratio (unless stated otherwise), acidified, and dried.

Before MS-analysis and fractionation, peptide samples were purified using either Empore C18 (Octadecyl) resin material (3M Empore) or tC18 SepPak (50mg, Waters). Material was activated by incubation with Methanol for 5 min, followed by washing each with 70% acetonitrile/0.1% TFA and 5% acetonitrile/0.1% TFA. Samples were resuspended in 5% acetonitrile, 0.1% TFA and loaded to resin material. Peptides were washed with 5% acetonitrile/0.1% TFA and eluted with 70% acetonitrile (ACN). Samples were dried for future use.

#### 2.10.2. High-pH Reverse Phase Fractionation

For high-pH reversed phase fractionation on the Dionex HPLC, 500 µg of pooled and purified TMT-labelled samples were resuspended in 10 mM ammonium-bicarbonate (ABC), 5% ACN, and separated on a 250 mm long C18 column (Aeris Peptide XB-C18, 4.6 mm ID, 2.6 µm particle size; Phenomenex, Aschaffenburg, Germany) using a multistep gradient from 100% Solvent A (5% ACN, 10 mM ABC in water) to 60% Solvent B (90% ACN, 10 mM ABC in water) over 70 min. Eluted peptides were collected every 45 s into a total of 96 fractions, which were cross-concatenated into 12 fractions and dried.

#### 2.10.3. LC-MS^3^

Peptides were resuspended in 0.1% FA and separated on an easy nLC 1200 (ThermoFisher Scientific) and a 22 cm long, 75 µm ID fused-silica column, which has been packed in house with 1.9 µm C18 particles (ReproSil-Pur, Dr. Maisch, Ammerbruch-Entringen, Germany), and kept at 45 °C using an integrated column oven (Sonation). Peptides were eluted by a non-linear gradient from 5–38% acetonitrile over 120 min and directly sprayed into a Fusion Lumos mass spectrometer equipped with a nanoFlex ion source (ThermoFisher Scientific) at a spray voltage of 2.6 kV. Full scan MS spectra (350–1400 m/z) were acquired at a resolution of 120,000 at m/z 200, a maximum injection time of 100 ms and an AGC target value of 4× 105 charges. MS^2^ scans were performed for up to the 10 most intense ions in the IonTrap (Rapid), with an isolation window of 0.7 Th, a maximum injection time of 85 ms, and CID-fragmented using a collision energy of 35% for 10 ms. SPS-MS^3^ was performed on the 10 most intense MS^2^ fragment ions with an isolation window of 0.7 Th (MS1) and 2 m/z (MS2). Ions were fragmented using HCD with a normalized collision energy of 65 and analyzed in the Orbitrap with a resolution setting of 50,000 at m/z 200, scan range of 110–500 m/z, AGC target value of 1 × 10^5^ and a maximum injection time of 86 ms. Dynamic exclusion was set to 45 s to minimize repeated sequencing of already acquired precursors. Raw filed data were processed with Proteome Discoverer 2.2 software and searched against the mouse SwissProt reference database, including isoforms (2018-12-10) and common contaminants. PSMs were filtered for a co-isolation threshold of 50% and an average reporter S/N of at least 10. For quantification, only unique peptides were considered, and data was corrected for impurities according to Lot-number and normalized using the total peptide amount to account for unequal loading in the TMT multiplex.

The mass spectrometry proteomics data have been deposited to the ProteomeXchange Consortium via the PRIDE partner repository (Proteomics Identifications Database) with the dataset identifier PXD019836 [34].

### 2.11. Western Blot

For Western Blot analysis, 20 to 30 µg of protein of treated bone marrow-derived macrophages were separated by SDS-polyacrylamide gel electrophoresis (4% stacking gel, 12% running gel). Blotting of the proteins was performed using the Trans-Blot^®^Turbo™ Transfer System (BioRad, Hercules, CA, USA) and a nictrocellulose membrane (Sigma-Aldrich, St. Louis, Missouri, USA).

Afterwards, the membrane was blocked with 5% milk powder in TN buffer for 2 h following antibody incubation over night at 4 °C: p-ERK 1:1000 (4377S, Cell Signaling, Frankfurt am Main, Germany), ERK 1:1000 (sc-1647, Santa Cruz Biotechnology, Dallas, TX, USA), p-p38 1:1000 (9211, Cell Signaling, Frankfurt am Main, Germany), p-38 1:1000 (9212, Cell Signaling), ROCK-1 1:500 (ab45171, abcam), ROCK-2 1:500 (ab71598, abcam,), Transgelin 1:1000 (NB600-507, novusbio), PTK7 1:500 (AF4499, R&D Systems), MMP9 1:1000 (AF909, R&D Systems), ITGA2B 1:1000 (orb376331, biorbyt, Cambridge, UK), CXCR4 1:1000 (BS-1011R, biossantibodies, Woburn, MA, USA), ELMO-1 (ab2239, abcam). As loading controls, either HSP90 1:1000 (sc-13119, Santa Cruz, Dallas, TX, USA) or GAPDH 1:1000 (ab8245, abcam) were used. Detection of proteins was performed with HRP-coupled secondary antibodies in a dilution of 1:5000 (ab97110, abcam; A9169-2ML and A9044-2ML, Sigma-Aldrich) and an ECL reagent (Thermo Scientific) [35]. Quantification of protein amount was determined by densitometrical analysis with GelAnalyzer 2010 software [36].

### 2.12. Fluorescence-Activated Cell Sorting (FACS) Analysis

Single-cell suspension of ipsi- and contra-lateral nerves, spinal cord and L4–L6 DRGs were generated by chopping the nerves into small pieces, mincing the spinal cord and then incubating all the tissues in DMEM (Invitrogen) with 3 mg/mL collagenase (Sigma) and 1 µL/mL DNase (Promega) (30 min, 37 °C, 95% O_2_) with shaking the samples every 10 min. Afterwards, DMEM with 10% FCS was added to the samples. Samples were then transferred through a 70 µm filter. The filter was washed with 0.5% BSA (Sigma) in PBS. Afterwards, cells were centrifuged (400× *g*, 5 min) and washed with 0.5% BSA in PBS. Single-cell suspensions were blocked with FcR blocking reagent (Miltenyi Biotec, Bergisch Gladbach, Germany) in 0.5% PBS-BSA for 20 min, stained with fluorochrome-conjugated antibodies and analyzed on a LSR II/Fortessa flow cytometer or sorted using a FACS Aria III cell sorter (BD Biosciences, Franklin Lakes, NJ, USA). Data were analyzed using FlowJo V10 (TreeStar). All antibodies and secondary reagents were titrated to determine optimal concentrations. Comp-Beads (BD) were used for single-color compensation to create multicolor compensation matrices. For gating, fluorescence minus one controls were used. The instrument calibration was controlled daily using Cytometer Setup and Tracking beads (BD Biosciences). For characterization of immune cell subsets in nerve, DRGs or spinal cord, the following antibodies were used: anti-CD3-PE-CF594, anti-CD4-V500, anti-CD8-BV650, anti-CD11b-BV605, anti-CD11c-AlexaFluor700, anti-CD19-APC-H7, anti-Ly6C-Per-CP-Cy5.5, anti-NK1.1 PE (BD Biosciences), anti-CD45-Vio-Blu, anti-MHC-II-APC (Miltenyi Biotec), anti-F4/80-PE-Cy7, anti-GITR-FITC and anti-Ly6G-APC-Cy7 (BioLegend, San Diego, CA, USA) [37].

### 2.13. Liquid Chromatography-Tandem Mass Spectrometry (LC-MS/MS) for the Determination of Lipid Mediators

The following reference substances as well as their deuterated analogues used as internal standards were purchased from Cayman Chemical (Ann Arbor, USA): 9(10)- and 12(13)-EpOME, (epoxyoactadecenoid acid) 9,10- and 12,13—DiHOME dihydroxyoactedecenoic acids), 9- and 13-HODE, (hydroxyoctadecadecenoic acis), 5,6-, 8,9-, 11,12- and 14,15-EpETrE (EET, epoxyeicosatrienoic acids), 5,6-, 8,9-, 11,12- and 14,15-DiHETrE (DHET, dihydroxyeicosatrienoic acids), 17(18)-EpETrE (EEQ, epoxyeicosatetraenoic acid) and 19(20)-EpDPA (EDP, epoxydocosapentaenoic acid).

After dissecting the contralateral and ipsilateral sides of sciatic nerve, the L4–L6 dorsal root ganglia and the spinal cord from euthanized animals, the samples were directly frozen in liquid nitrogen. The tissue weight was determined, and lipid quantification was performed as described previously [16,38]. Briefly, tissue samples were homogenized using a swing mill and analytes were extracted from homogenates using liquid–liquid extraction with ethyl acetate after spiking with a mixture of the deuterated internal standards. After liquid–liquid extraction, combined organic phases were removed at a temperature of 45 °C under a gentle stream of nitrogen. The residues were resuspended in 50 μL of methanol/water/BHT (50:50:10^−4^, *v*/*v*/*v*), then centrifuged for 2 min at 10,000× *g* and transferred to glass vials (Ziemer GmbH, Langenwehe, Germany).

For calibration, PBS samples were spiked with working solutions of the analytes (prepared in methanol/BHT (100:0.1 (*v*/*v*)) and processed as described for the homogenates.

The LC-MS/MS system consisted of a triple quadrupole tandem mass spectrometer QTRAP 5500 (Sciex, Darmstadt, Germany) equipped with a Turbo-V source operating in negative electrospray ionization mode, an Agilent 1200 binary HPLC pump and degasser (Agilent, Waldbronn, Germany) and a HTC Pal autosampler (CTC analytics, Zwingen, Switzerland).

Chromatographic separation of the lipids was performed using a Gemini NX C18 column and precolumn (150 × 2 mm inner diameter, 5 µm particle size and 110 Å pore size; Phenomenex) and a linear gradient using a flow rate of 0.5 mL/min in a total run time of 17.5 min. Thereby, the gradient changed from 85% mobile phase A (water:ammonia 100:0.05, *v*/*v*), to 10% A and 90% mobile phase B (acetonitrile:ammonia 100: 0.05, *v*/*v*). The conditions were held for 1 min until the mobile phase shifted back to 85% A. These conditions were maintained for 4 min to re-equilibrate the column.

Analyst software version 1.6.3 (Sciex) was used for data acquisition, while further quantification was performed with Multiquant Software version 3.0.2 (Sciex) using the internal standard method (isotope-dilution mass spectrometry). Calibration curves were calculated by linear regression with 1/concentration weighting.

### 2.14. Data Analysis and Statistics

All data are presented as mean ± SEM. Determination of statistically significant differences in all behavioral experiments was conducted with two-way analysis of variance (ANOVA) followed by post hoc Bonferroni correction using GraphPad Prism 7. For in vitro experiments comparing only two groups, Student’s *t* test was carried out with Welch’s correction, and for comparing more than two groups, one-way ANOVA was used. The comparison of more groups with different conditions, including behavioral experiments, was performed with two-way ANOVA followed by post hoc Bonferroni or Holm-Sidak correction. A *p-*value of <0.05 was considered statistically significant. Lipidomic data were tested for normal distribution with the Shapiro–Wilk Test using GraphPad Prism 7 and all datasets passed the test (W > W_α_; α = 0.05).

## 3. Results

### 3.1. Loss of G2A Alleviates Mechanical Hypersensitivity and Alters Lipid Signaling

We previously showed that activation of G2A in sensory neurons leads to an increased mechanical pain hypersensitivity during oxaliplatin-induced neuropathic pain due to sensitization of TRPV1 [16]. However, G2A is also strongly expressed in immune cells, such as macrophages, neutrophils and T-cells [18,39]. We therefore wanted to investigate the role of G2A in a neuropathic pain model with a strong inflammatory component. Since peripheral nerve injury-induced neuropathic pain is characterized by a strong immune cell infiltration from day 4, ongoing and lasting for over 21 days [40,41], we chose the robust spared nerve injury (SNI) model [27]. After performing the SNI surgery, we analyzed the paw withdrawal reflex of the mice for seven days. We found that G2A-deficiency resulted in a significant decrease of mechanical hypersensitivity from starting at day 2 compared with wild-type mice (Figure 1A).

At the site of injury, immune cells are recruited and activated, stimulating peripheral sensory neurons to release endogenous factors, such as lipid mediators [6,10,42]. Since G2A is a lipid receptor, we investigated whether or not the levels of lipids known to activate G2A as well as their metabolites were altered during SNI-induced neuropathic pain in nervous tissue. Therefore, we performed LC-MS/MS-based targeted analysis. We observed that the linoleic acid metabolites 9- and 13-HODE were increased at the injured nerves in wild-type mice 7 days after the SNI surgery (Figure 1C,E), both of which are endogenous agonists of G2A (Figure 1B). Moreover, we could detect a strong increase of the lipid 12,13-epoxyoctadecenoic acid (EpOME) in the injured ipsilateral site of the nerve, which was recently identified as weak G2A agonist (Figure 1G) [43]. In contrast, G2A-deficient mice did not show any difference between the treated (ipsilateral) and untreated sites (contralateral) regarding the different lipid concentrations (Figure 1D,F,H,J). In DRG tissue of wild-type mice, the DiHOMEs showed an increased level at the ipsilateral site, indicating a strong involvement of soluble epoxide hydroxylase (sEH) in the DRGs (Figure 1I) [44]. Similar results were observed 24 h after zymosan-induced inflammatory pain and also 7 days after CFA-induced inflammation in DRGs [45]. This indicates a role for 12,13-DiHOME in pain processing since it also showed decreased amounts in the sciatic nerve in G2A-deficient mice (Figure 1H).

### 3.2. G2A-Dependent Alterations of Immune Cell Recruitment and Cytokine Synthesis in Nociceptive Tissue

Since G2A agonists were upregulated in the different nociceptive tissues, we next analyzed the number of immune cells along the stations of nociceptive processing. Using multiparameter FACS analysis, we observed an infiltration of various immune cells in general in wild-type mice at the site of injury of the sciatic nerve (SN) 7 days after SNI surgery (Figure 2A). The infiltration of immune cells to the DRGs and the spinal cord after nerve injury has been demonstrated before and usually peaks around days 5–10 after surgery [46,47,48,49]. Based on these observations, we considered 7 days after surgery an appropriate timepoint to investigate immune cell infiltration into nervous tissue.

Interestingly, G2A-deficient mice showed a 5- to 10-fold reduced number of immune cells (CD45-positive cells) at the site of injury in SN (Figure 2A–C). However, the strongest differences were observed in the number of macrophages and neutrophils in sciatic nerve (SN) (Figure 2B). Here, the number of macrophages was 12-fold lower at the site of injury in G2A-deficient mice than in wild-type mice, which could be confirmed with immunohistochemistry staining for F4/80 and CD11b (Figure 2D,E). Likewise, the number of neutrophils was 7-fold lower in G2A-deficient mice (Figure 2B). Next to this, the number of dendritic cells (DC), CD11b+ NK cells, NK cells and T cells revealed a decrease in G2A-deficient mice as well, in ipsilateral SN (Figure 2C).

In L4–L6-DRGs, the number of infiltrated immune cells was smaller than in SN (Figure 2A–H). As the FACS analysis revealed, the number of infiltrated CD45^+^ immune cells was decreased in G2A-deficient mice 7 days after SNI surgery (Figure 2F). Next to this, only the macrophages and neutrophils showed a significantly decreased number in G2A-deficient mice (Figure 2G). However, the number of macrophages was not as strongly decreased as in the sciatic nerve (Figure 2B,G). Moreover, the number of neutrophils was also decreased in DRGs, compared to SN in G2A-deficient mice 7 days after SNI (Figure 2B,G). However, the other immune cell types showed no differences in DRGs in both genotypes (Figure 2H).

Interestingly, the infiltration of immune cells into the spinal cord was in total 10-fold lower than in the sciatic nerve and 3.5-fold smaller than in DRGs (Figure 2A,F,I). Nevertheless, the CD45^+^ immune cells in spinal cord also showed a decreased number in G2A-deficient mice 7 days after SNI (Figure 2I). However, only the number of macrophages were significantly decreased in spinal cord in G2A-deficient mice, compared to wild-type mice (Figure 2J).

There was no difference observed comparing astrocyte activation between the genotypes 7 days after SNI, which is known to be characteristic in neuropathic pain [50] (Figure 2I–K, Appendix A).

In conclusion, we saw differences in the number of immune cells, especially macrophages, mainly at the site of injury. The differences in immune cell infiltration between the genotypes were smaller in DRGs and in the spinal cord (Figure 2).

The reduced number of immune cells, especially at the site of injury, lead us to the hypothesis that the levels of cytokines, chemokines and growth factors, known for modulating neuropathic pain [9,11], may also be altered in G2A-deficient mice after SNI surgery. We therefore performed an unbiased screen of growth factors, chemokines and cytokines, which are known for their role in inflammation, which was previously shown in a SNI model [41] (Figure 3A). In this screen, we observed decreased concentrations of tumor necrosis factor alpha (TNFα) and interleukin 6 (IL-6) in ipsilateral sciatic nerves of G2A-deficient mice compared to wild-type mice (Figure 3B,C). These two signaling molecules have been connected with persistent pain states before and are known mediators of both peripheral and central sensitization [51]. The cytokines GM-CSF, IFNγ, IL-4, IL-10, MCP-1 and MIG were not detectable in any tissue measured (data not shown). Additionally, the concentrations of nerve growth factor (NGF), which is a very important factor released upon injury and inflammation contributing to pain initiation [12,14], as well as the anti-inflammatory cytokine TGFβ, did not show any differences when comparing wild-type and G2A-deficient mice (Figure 3D,E). NGF expression is known to be increased by pro-inflammatory cytokine IL-1β [11,14], but IL-1β also showed no difference between both genotypes at the ipsilateral site of injured nerves, nor was a general difference in G2A expression detectable in DRGs after surgery (Appendix A).

In L4–L6 DRG, we observed a decreased amount of the cytokine IL-12 in G2A-deficient mice (Figure 3A). However, the expression of either IL-12a (data not shown) or IL-12b remained unaltered (Appendix A), indicating that neuronal G2A is unaffected by SNI surgery. Oxidative stress is an important contributor to nerve injury-induced neuropathic pain. However, oxidative stress markers such as inducible NO synthase (iNOS), NADPH oxidase (Nox)-2 and Nox-4 did not show any alterations in mRNA expression in wild-type and G2A-deficient mice (Appendix A). In contrast, the established neuronal stress marker ATF3 (activating transcription factor 3) was increased 7 days after SNI (Appendix A).

To investigate whether or not these differences between the genotypes are caused by the surgery itself, we also measured immune cell number, cytokine, chemokine and growth factor concentrations between the two genotypes 1 day after SNI surgery. At this time point, no differences were found between the genotypes in ipsilateral sites, either in the number of immune cells or the amount of various cyto- and chemo-kines in SN, DRGs or SC (Appendix A). These data suggest that surgery-induced acute recruitment of immune cells is not G2A-dependent and that the differences between the genotypes require a neuropathic component that develops several days after nerve-injury.

### 3.3. Migration of Macrophages is Impaired by Loss of G2A Receptor

Since the number of macrophages was dramatically decreased in G2A-deficient mice in all three investigated tissues (sciatic nerve, DRGs and spinal cord), we hypothesized that G2A-deficiency affects the immunomodulatory and migratory properties of macrophages [22]. Therefore, we performed a global proteome analysis using a vanguard mass-spectrometry-based approach [33] using bone marrow-derived macrophages (BMDM) of wild-type and G2A. The BMDMs were stimulated with the G2A agonist 9-HODE (1 µM) for 24 h prior to the proteomic analysis. This concentration is in range of the EC_50_ of 9-HODE for both the human and the murine G2A, as previously shown [16,21]. The stimulation of BMDMs with 9-HODE induced a minor, but significant increase of G2A mRNA expression (Figure 4A).

Global proteome analysis revealed significant changes in protein levels in wild-type mice after G2A activation through 9-HODE (Figure 4B). Among the strongest upregulated proteins were protein tyrosine kinase 7 (PTK7), catenin alpha-1 (Ctnna1), alpha actinin 1 (Actn1), myeloid differentiation primary response protein (Myd88), transgelin (Tagln) and phosphoinositide 3-kinase (PI3K) subunits, which are known to play central roles in migration (Figure 4B and Figure 5A, Appendix A) [52,53,54,55,56,57]. Proteins like caspase 8, cathepsines and Gadd45 were the proteins with the strongest downregulation (Figure 3C, Appendix A). These proteins are known to play a central role during apoptosis [58,59]. Interestingly, proteins concerning hematopoiesis were rather downregulated, including CD38 (Figure 4C, Appendix A), which is in line with previous observations showing that G2A receptor affects hematopoiesis in vivo [43].

### 3.4. Is Macrophage Migration Affected by MMP9 Regulation?

Among the strongest regulated proteins known to mediate migration were proteins connected to the Toll-like receptor 4 (TLR4) signaling pathway, such as MyD88, PI3K and Dock2 (Figure 5B,C), as well as matrix-metalloproteinase 9 (MMP9) (Figure 5D,E). This finding was especially surprising, since MMP9 is a key molecule in macrophage migration in the tissues, due to its ability to destroy extracellular matrix [60].

However, MMP9 downregulation in BMDMs after 24 h of G2A-activation through 9-HODE was confirmed by Western blot and with ELISA (Figure 5E,F). Interestingly, in a time-dependent stimulation study, MMP9 secretion was significantly increased 4, 6 and 10 h after 9-HODE stimulation (Figure 5G). Whereas, in BMDM lysates, MMP9 expression decreased over time, which is in line with an increased secretion (Figure 5F). Other pro-migratory proteins, such as MAP-kinases, ROCK, Ras and Rac were not affected by 9-HODE stimulation nor were there any differences between the genotypes (Appendix A, Appendix A).

## 4. Discussion

In this study, we showed that loss of G2A alleviated mechanical hypersensitivity in acute nerve injury-induced peripheral neuropathic pain. As the results suggest, this is likely due to an altered neuroimmune response accompanied by a reduced immune cell infiltration to the injured nerve. Thereby, the concentrations of established inflammatory mediators released by immune cells, such as IL-6 and TNFα, but also of oxidized lipid mediators like 9- and 13-HODE (Figure 1, Figure 2 and Figure 3) are markedly decreased at the injured sciatic nerve of G2A-deficient mice.

For the sensitizing effects of 9-HODE, its receptor G2A seems to be required, which is induced during cellular stress and DNA damage [16,61]. Since nerve injury is stress for cells and tissues, it is not surprising that the expression of the known neuronal stress marker ATF3 was increased 7 days after SNI (Appendix A) [62]. However, stress markers were not differentially expressed between the genotypes, indicating no effect on cellular stress due to G2A-deficiency.

G2A-deficient mice showed less mechanical hypersensitivity in an oxaliplatin-induced neuropathic pain model, suggesting an anti-nociceptive effect of G2A-deficiency [16]. Similar results were observed after peripheral nerve injury-induced neuropathic pain during the first 7 days (Figure 1). After nerve injury, we did not see a difference in the expression levels of G2A in the DRGs of wild-type mice, compared with untreated mice (Appendix A). However, we observed an increased expression of G2A in BMDMs after stimulation with 9-HODE (Figure 4A). This suggests different actions and pathways involved upon G2A-signaling in various tissues and cell types in pain and inflammation, indicating that the neuronal G2A population does not contribute to SNI-induced neuropathic pain.

Nerve injury-induced neuropathic pain is characterized by a strong inflammatory component, that involves immune cell migration to the injured site of the peripheral nerve [63]. Previous studies have shown a recruitment of immune cells into the spinal cord, beginning 3 days after SNI, as well as the upregulation of pro-inflammatory factors, such as IL-6, lasting for over 21 days [41,47,64]. Here, we demonstrated that G2A-deficient mice showed a strongly reduced infiltration of immune cells at the site of injury. Consistent with our findings, earlier studies also showed a reduced number of macrophages at the inflammatory site during peripheral inflammatory pain [22]. Interestingly, in dextran sulfate sodium (DSS)-induced colitis, a reduced number of T cells was observed in G2A-deficient mice, confirming our data of the sciatic nerve after nerve injury (Figure 2C) [65]. Overall, these results suggest that G2A-signaling may depend on the type and location of inflammation as well as release and distribution of signaling lipids in the respective tissues [66].

The G2A receptor seems to influence migration of macrophages since it is able to change cell morphology and cytoskeleton structure [23]. These observations are in line with our data. The most prominent group of proteins being upregulated in macrophages after G2A activation are migratory proteins, as well as proteins responsible for cytoskeleton remodeling. Here, after 9-HODE stimulation, we could show upregulated G2A expression in macrophages but not in DRGs (Figure 4A, Appendix A).

MMP9 is known to play a crucial role in the development of neuropathic pain and inflammation [67]. Indeed, we could observe that MMP9 release is transiently increased during the first 10 h after 9-HODE-induced G2A activation in BMDMs. However, 24 h after G2A activation, we observed decreased MMP9 levels in macrophage cell lysates. This indicates that MMP9 release is increased within the first hours after G2A activation but is saturated 24 h after G2A activation, leaving low intracellular concentrations at this timepoint.

Moreover, we found MyD88, transgelin, PI3K-components and Akt1 to be strongly upregulated in our proteome screen 24 h after G2A activation (Figure 5), which is in good agreement with earlier studies [64]. These proteins all belong to the TLR4 signaling pathway, that is known to promote leukocyte migration and cause release of TNFα and IL-6, both of which we found strongly reduced in G2A-deficient mice (Figure 3B,C) [68,69,70]. MMP9 is also a downstream target of TLR4 signaling [71]. The transiently increased secretion of MMP9 during the first 10 h of G2A activation is in line with pro-migratory events through TLR4 signaling. According to our data, we therefore propose that G2A activation in macrophages initiates MyD88-PI3K-AKT signaling, transient MMP9 release, as well as TNFα and IL-6 expression and a potential crosstalk with PTK7-Wnt signaling to initiate cytoskeleton remodeling and migration. This is also in good agreement with the data of the proteome screen, wherein MyD88 was not differentially regulated in G2A-deficient bone marrow-derived macrophages (Figure 5B).

However, the study faces several limitations since the proteome analysis data was performed with bone marrow-derived macrophages stimulated in vitro with 9-HODE. Moreover, it is unclear whether these results can be translated to ameliorate neuropathic pain in patients.

Thus, we showed that the immune response is strongly reduced in G2A-deficient mice through a strong reduction of migrated immune cells and reduced concentrations of proinflammatory cytokines, like TNFα and IL-6, at the site of injury (Figure 3B,C).

It is known that IL-1β, IFN-γ, IL-17, IL-6 and TNFα are increased in nervous tissue in animal models of neuropathic pain, but also in cerebrospinal fluid and blood of patients with neuropathic pain [72,73]. Of those, especially IL-6 and TNFα seem to play crucial roles in the initiation of neuropathic pain, in activation of macrophages and in pain processing through sensitization of the TRPV1 channel [74,75]. However, the heterogeneity of neuropathic pain syndromes in patients, the individual differences in disease progression and the differences in investigated liquor or tissue makes it difficult to compare animal and patient studies and to identify specific markers for neuropathic pain.

In the present study, G2A-deficiency resulted in a reduction of IL-6 and TNFα (Figure 3B,C). The decreased amount is likely due to reduced immune cell infiltrates at the site of injury. Furthermore, it was shown that TNFα, IL-6 and IL-12 were increased in G2A-deficient macrophages in atherosclerosis [76]. However, this study was performed in double knockout mice for G2A and ApoE [76] and the strongly reduced macrophage infiltrates observed in our system may easily compensate for a relative increase in inflammatory cytokine production by individual cells. Such different observations may also be explained by the pleiotropic effects of G2A in different inflammatory environments. Thus, Kern and colleagues did not find any difference in the amount of 9-HODE at the inflammatory site, whereas in an earlier study, 9-HODE induced production of IL-6, indicating an important role of the 9-HODE–G2A axis during inflammatory processes [22,77].

## 5. Conclusions

We could demonstrate that G2A is a crucial component in the initiation of inflammation after nerve injury and thus contributes to neuropathic pain. Furthermore, we showed that G2A-deficiency leads to a strong reduction of macrophage migration to the location of the injured nerve, probably via reduction of TLR4-MyD88-PI3K-AKT and MMP9 signaling and a potential crosstalk with PTK7-Wnt signaling, leading to decreased TNFα and IL-6 levels at the site of injury.

The treatment of neuropathic pain is complex, and so far, sufficient medication is not available [78]. Clinical trials dealing with TNFα or IL-6 inhibitors did not show conclusive and persuasive results [73]. Thus, G2A may represent a promising new target, by inhibitors or antibodies against G2A, and thereby preventing leukocyte migration and peripheral sensitization, especially at the onset of nerve injury-induced neuropathic pain in patients. While, generally, off-target effects of systemic G2A-inhibitors cannot be excluded, they should mainly target G2A receptors in leukocytes where the receptor shows by far its highest expression [23] and other cell types should not be strongly affected by these substances.

However, further preclinical studies are required to assess potential side effects of inhibiting G2A in patients. G2A-inhibitors may have a profile of side effects that is similar to immunosuppressants, which is mainly characterized by a higher risk of infections. We therefore consider treatment with G2A-inhibitors to be carefully monitored and to be restricted to the onset of nerve injury-induced neuropathic pain.

## Figures and Tables

**Figure 1 cells-09-01740-f001:**
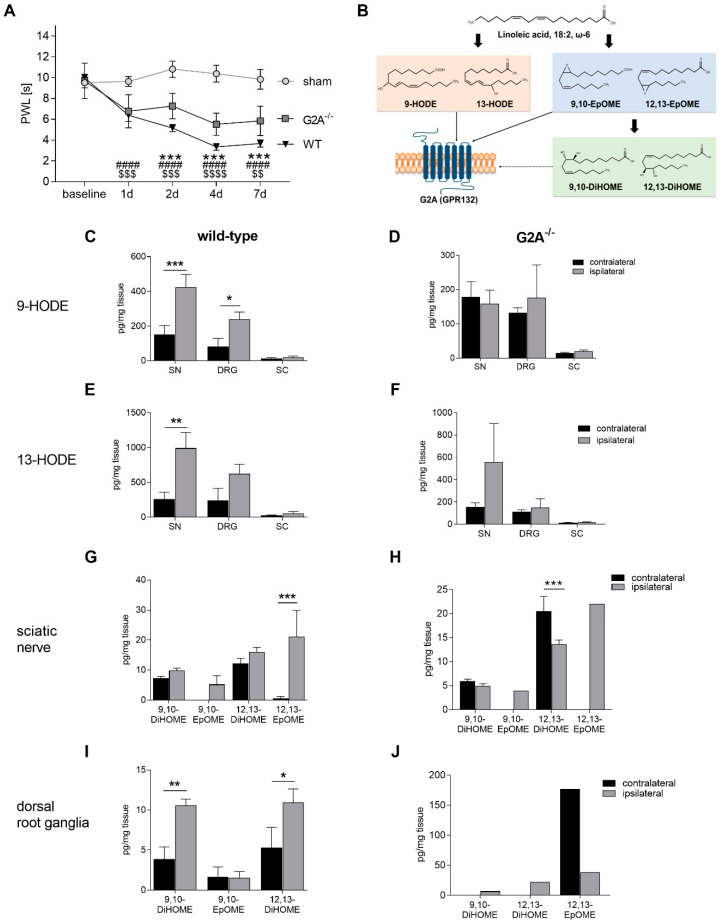
GPR132 (G2A)-deficiency leads to reduced mechanical hypersensitivity and the secretion of oxidized lipids is affected in wild-type mice. (**A**) Paw withdraw latency (PWL) of ipsilateral site of wild-type mice (WT, triangle), G2A-deficient mice (G2A^−/−^, squares) and sham-treated mice (circles) after spared-nerve injury (SNI) surgery. n = 10 animals, male and female, * *p* < 0.05, ** *p* < 0.01, *** *p* < 0.005, **** *p* < 0.001 (WT vs. G2A^−/−^); ### *p* < 0.005, #### *p* < 0.0001 (WT vs. sham); $$ *p* < 0.01, $$$ *p* < 0.005, $$$$*p* < 0.0001 (G2A^−/−^ vs. sham). Statistics were performed with two-way Analysis of variance (ANOVA) with Bonferroni correction. (**B**) Schematic depiction of the oxidative linoleic acid pathway. (**C**,**D**) Concentrations of 9- hydroxyoctadecadienoic acid (HODE) in sciatic nerve (SN), L4–L6-dorsal root ganglia (DRG) and spinal cord (SC) in wild-type (**C**) and G2A-deficient (G2A^−/−^) (**D**) mice 7 days after SNI surgery. (**E**, **F**) Concentrations of 13-HODE in sciatic nerve (SN), dorsal root ganglia (DRG) and spinal cord (SC) in wild-type (**E**) and G2A-deficient (G2A^−/−^) (**F**) mice 7 days after SNI surgery. (**G**–**J**) Concentrations of epoxyoctadecenoic acids (EpOMEs) and dihydroxyocatadecenoic acids (DiHOMEs) in SN (**G**,**H**) and DRG (**I**,**J**) in wild-type (**G**,**I**) and G2A^−/−^ (**H**,**J**) mice 7 days after SNI surgery. Black represents untreated site (contralateral). Grey represents treated site (ipsilateral) in the respective tissues from n = 5 mice per group, male and female. Data represent mean ± standard error ofmean (SEM). * *p* < 0.05, ** *p* < 0.01, *** *p* < 0.005; Two-way ANOVA with the Holm-Sidak method.

**Figure 2 cells-09-01740-f002:**
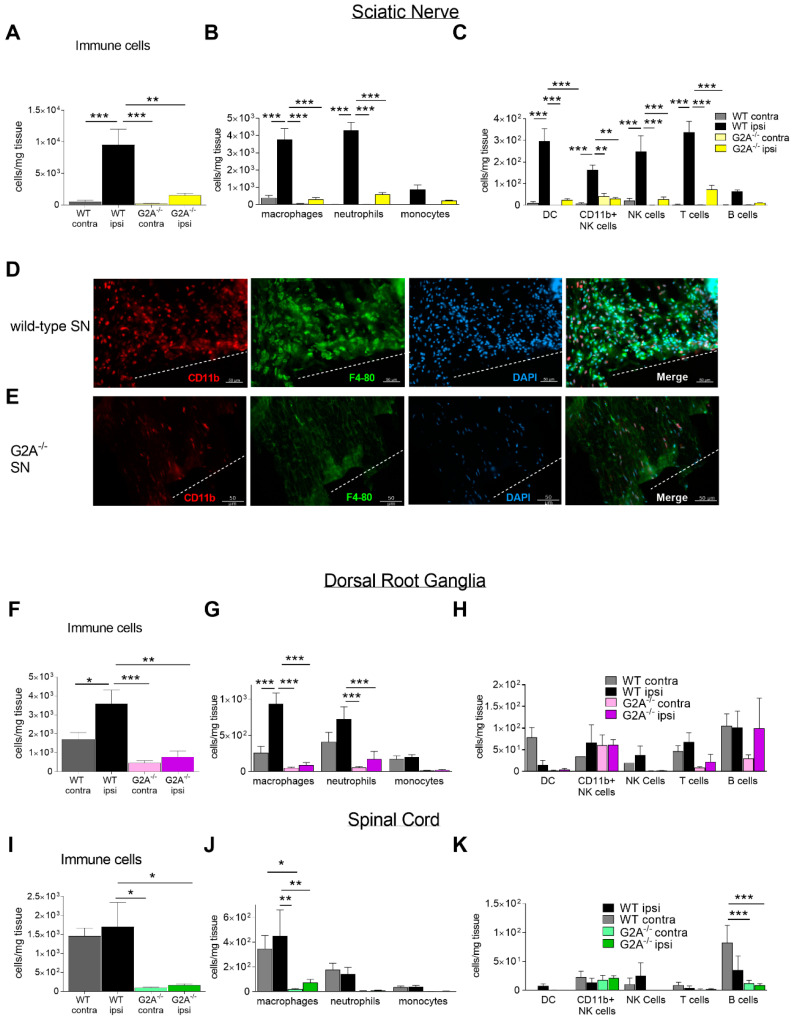
Reduced number of immune cells in G2A-deficient mice 7 days after SNI surgery. (**A**) Total immune cell number at the ipsi- and contra-lateral site of sciatic nerve (SN) (CD45^+^). (**B**,**C**) Number of different types of immune cells in the ipsi- and contra-lateral site of the sciatic nerve. (**D**,**E**) Immunohistochemical staining of macrophages (F4/80, CD11b) at the injured sciatic nerve (SN) in wild-type (**D**) and G2A-deficient mice (**E**) 7 days after SNI. Dashed lines indicate sites of injury. (**F**) Total immune cell number in L4-L6-DRGs at the ipsi- and contra-lateral site of sciatic nerve. (**G,H**) Number of different types of immune cells in the ipsi- and contra-lateral site of L4–L6-DRGs 7 days after SNI surgery. (**I**) Total immune cell number in the dorsal spinal cord receiving input from spinal cord comparing the ipsi- and contra-lateral site. (**J**,**K**) Number of different types of immune cells in the dorsal spinal cord section receiving input from L4–L6-DRGs comparing the ipsi- and contra-lateral site 7 days after SNI surgery. Data were obtained from *n* = 5 animals per group, male and female. Ipsilateral site of WT mice is shown in black. Ipsilateral site in SN is shown in yellow, in DRG in purple and in SC in green. Contralateral site of WT mice is depicted in dark grey. Contralateral site of G2A^-/-^ mice is shown in light grey. Neutrophils (CD45^+^, Ly6G^+^, CD11b^+^), macrophages (CD45^+^, Ly6G^−^, CD11b^+^, F4-80^+^, Ly6C^−^), monocytes (CD45^+^, Ly6G^−^, CD11b^+^, F4-80^−^, Ly6C^+^), dendritic cells (DC; CD45^+^, Ly6G^−^, CD11b^+^, F4-80^−^, Ly6C^−^, CD11c^+^, MHCII^+^), CD11b^+^ NK cells (CD45^+^, Ly6G^−^, CD11b^+^, F4-80^−^, Ly6C^−^, NK1.1^+^), NK cells (CD45^+^, Ly6G^−^, CD11b^−^, F4-80^−^, Ly6C^−^, NK1.1^+^), T cells (CD45^+^, Ly6G^−^, CD11b^−^, F4-80^−^, CD3^+^, MHCII^−^), CD4 T cells (CD45^+^, Ly6G^−^, CD11b^−^, F4-80^−^, CD3^+^, MHCII^−^, CD4^+^), CD8 T cells (CD45^+^, Ly6G^−^, CD11b^-^, F4-80^−^, CD3^+^, MHCII^−^, CD8^+^), B cells (CD45^+^, Ly6G^−^, CD11b^−^, F4-80^−^, CD11c^−^, CD3^−^, MHCII^+^, CD19^+^). Data represents mean ± SEM. * *p* < 0.05, ** *p* < 0.01, *** *p* < 0.005, **** *p* < 0.001; Two-way ANOVA with Bonferroni correction.

**Figure 3 cells-09-01740-f003:**
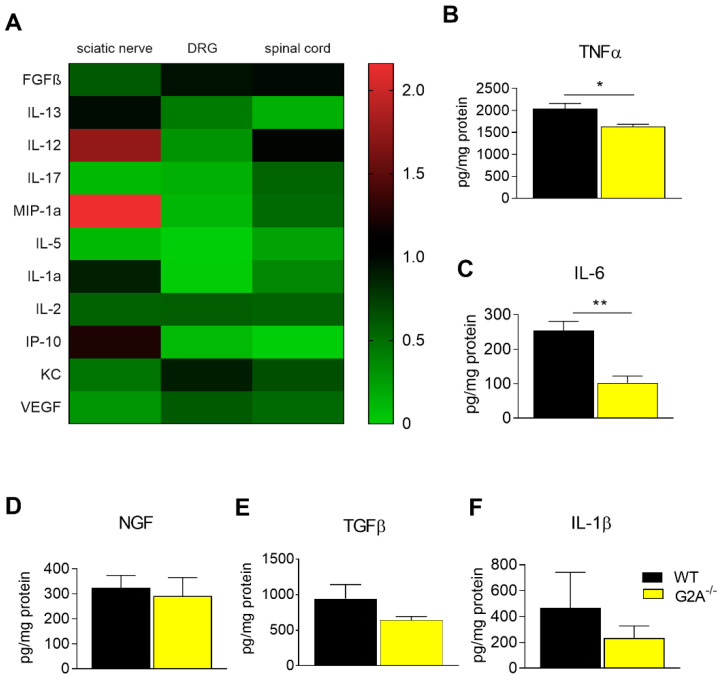
Reduced concentrations of inflammatory cytokines in G2A-deficient mice. (**A**) Heat map of cytokines levels in sciatic nerve, DRG and spinal cord at the ipsilateral site of wild-type and G2A-deficient mice 7 days after SNI. Data shown as ipsi-lateral of WT vs. ipsi-lateral of G2A-defcient mice measured with LUMINEX. (**B**–**F**) Concentrations of tumor necrosis factor α (TNFα) (**B**), interleukin 6 (IL-6) (**C**), nerve growth factor (NGF) (**D**), transforming growth factor β(TGFβ) (**E**) and IL-1β (**F**) in the ipsilateral site of sciatic nerve 7 days after SNI in wild-type (WT, black) and G2A-deficient mice (G2A^−/−^, yellow in sciatic nerve), measured with enzyme-linked immunosorbent assay (ELISA). Data represent mean ± SEM. * *p* < 0.05, ** *p* < 0.01, *** *p* < 0.005, **** *p* < 0.001 of n = 5 mice per group, male and female; one-way ANOVA.

**Figure 4 cells-09-01740-f004:**
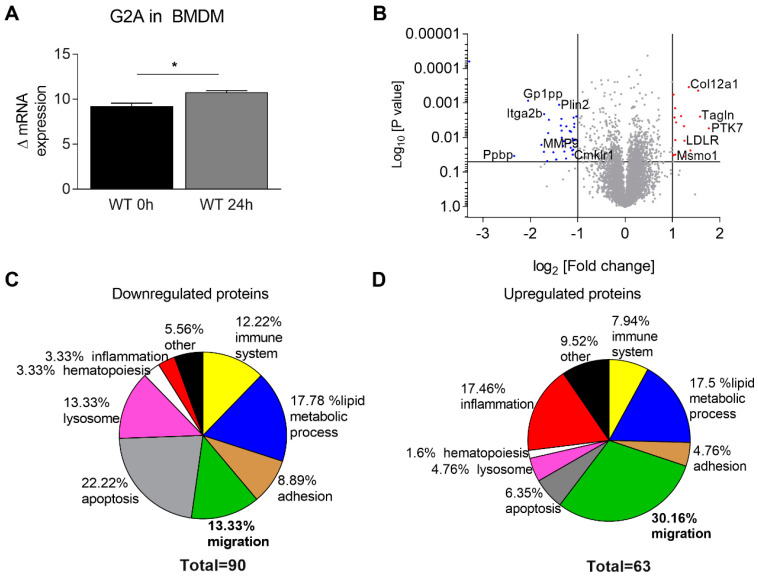
Protein regulation in wild-type bone marrow-derived macrophages (BMDMs) 24 h after stimulation with 9-HODE. (**A**) Relative mRNA expression of G2A receptor in BMDMs treated with 1 µM 9-HODE for 24 h. n = 6–10 animals per group, male. (**B**) Identification of >6.000 regulated proteins 24 h after stimulation with 1 µM 9-HODE in BMDMs. Fold change (FC) [log2] is plotted against *p*-values [Log10]. Significant downregulated proteins are depicted in blue, and red represents upregulated proteins. (**C**) Percentage of downregulated proteins 24 h after 1 µM 9-HODE stimulation compared to untreated BMDMs. (**D**) Percentage of upregulated proteins 24 h after 1 µM 9-HODE stimulation compared to untreated BMDMs, clustered in groups. yellow: immune system, blue: lipid metabolic process, brown: adhesion, green: migration, grey: apoptosis, pink: lysosome, red: inflammation, white: hematopoiesis, black: other. Data represents mean ± SEM. * *p* < 0.05. An unpaired one-tailed *t*-test was used for statistics.

**Figure 5 cells-09-01740-f005:**
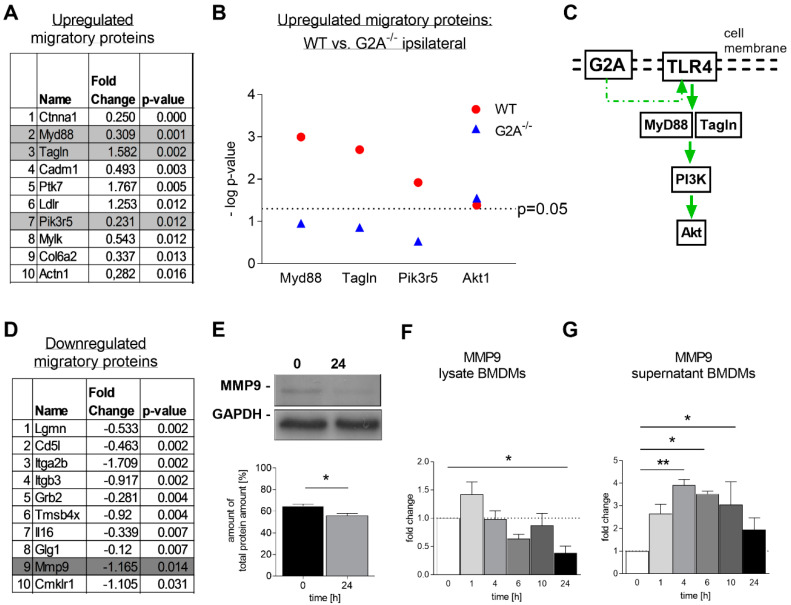
Transient increase of matrix-metalloproteinase 9 (MMP9) secretion and MMP9 downregulation in BMDMs 24 h after G2A activation. (**A**) List of the strongest upregulated migratory proteins in WT 24 h after 1 µM 9-HODE stimulation. (**B**) Comparison of the strongest upregulated migratory proteins in WT and G2A-deficient (G2A^−/−^) BMDMs regarding the toll-like receptor 4 (TLR4)-signaling pathway shown as -log *p*-value. (**C**) Scheme of possible G2A–TLR4 interaction and signaling in BMDMs. (**D**) List of the 10 strongest downregulated migratory proteins in WT 24 h after 1 µM 9-HODE stimulation. (**E**) Representative Western Blots of MMP9 expression and respective analyzed data in untreated and treated BMDMs with 1 µM 9-HODE. (**F**) MMP9 expression in BMDM lysates treated with 1 µM 9-HODE for different time points. (**G**) MMP9 secretion of BMDMs treated with 1 µM 9-HODE for different time points. Analyzed with ELISA. *n* = 6–10 male animals per group. Data represents mean ± SEM. * *p* < 0.05, ** *p* < 0.01. An unpaired one-tailed *t*-test and a One-way ANOVA were used.

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
