# Peer review of "The Lipid Receptor G2A (GPR132) Mediates Macrophage Migration in Nerve Injury-Induced Neuropathic Pain"

_cells, 2020, doi:10.3390/cells9071740_

Round 1
Reviewer 1 Report
The paper by Osthues et al. studies G2A receptors in traumatic nerve injury. The main effect contributing to less hypersensitivity seems to rely on reduce immune cell infiltration - specifically macrophages. Along with this, cytokines and proalgesic lipids are reduced. The paper provides an interesting link to previous work of the authors. I have the following major comments.
Major comments:
- The results of the macrophages stimulated in vitro are quite interesting. What is missing is the experimental link to the in vivo situation. Is the suspected pathway actually seen in WT and lost in KO? The authors should provide some data here to support the validity of bone marrow macrophages as a suitable model.
- Fig. 1: The header is misleading, because the lipids were measured in KO mice, but this is in S1. Please rather integrate this in Fig. 1. Why is there no error bar in Fig. S1D although it was n=5?
- Fig. 2: How clear is it, that the immune cells identified are really infiltrating rather than resident in the DRG and spinal cord. Is there really constant influx of immune cells into the spinal cord and the DRG in naive mice? Please discuss and revise these paragraphs.
- Fig. 3: The selection of cytokines is not easy to understand. Why are some of the results in SN/DRG shown and others not or only in the supplement? Were results from Luminex confirmed by ELISA? E.g. IL-12 is altered in the heat map but not confirmed by ELISA. Please provide all data in the main manuscript and make sure that major findings from the screen are confirmed.
- Fig. 4: Rational for using 1µM 9-HODE for in vitro stimulation? Sounds like a high concentration? Are there any data what tissue concentrations are compared to Fig. 1C?
Minor comments:
- Fig. 2: Please always keep the same order of the bars (ipsi/contra WT/KO mice). For clarity, the contralateral of the KOs mice could just be a lighter version of the color instead of very light grey (like the WTs).
- Fig. 2 legend: please add the markers used in FACS to identify the cells.
- Fig. 4: Please add WT in the header and then delete it in the rest of the figure legend.
- Fig. 5: The header is misleading: Everything was tested after 9-HODE stimulation? Then "after G2A activation" is not necessary.
- Fig. S3: The phrase SNI-treatment could easily be misunderstood, because SNI mice were not treated in addition. Statistics: multiple t-test? If more than 2 groups ANOVA was probably done.
- Methods: Please describe the proteins tested by Luminex in more detail.
Reviewer 2 Report
Please add the total number of animals used. Please add the sex of animals. If authors have followed, ARRIVE guidelines, it must be added.
Considering the time line of the study, is this a peripheral acute neuropathic model with dominant inflammatory imbalance? Is 7-day period enough to see changes? How fast those changes can return to baseline?
What is the difference to measure systemic biomarkers in blood and site of injury (tissue)? Correlated? positively or no correlation? In the discussion, authors have mentioned “…..but also in patients with neuropathic pain [68, 69]”. Levels of biomarkers in tissue or in systemic blood?
What is the time point for results in fig 2? 7day?
In n=5 mentioned in one of the figs, data were normally distributed to use parametric tests and presentation as mean or non-normal (to use median and IQ)? There is no normality check in statistic section.
In Fig 2 H is there any difference between the presented data?
What are the limitations of this study? How results can be translated in humans? What would be the potential side effects?
Authors have suggested “G2A may represent a promising new target, by preventing leukocyte migration and peripheral sensitization especially at the onset of nerve-injury induced neuropathic pain”. How the targeting could be achieved in humans?
Round 2
Reviewer 1 Report
All my comments were sufficiently addressed except for Comment 1. This reviewer understands that it is impossible to address this within 10 d.